# A Data-Based Approach Using a Multi-Group SIR Model with Fuzzy Subsets: Application to the COVID-19 Simulation in the Islands of Guadeloupe

**DOI:** 10.3390/biology10100991

**Published:** 2021-09-30

**Authors:** Sébastien Regis, Silvere P. Nuiro, Woody Merat, Andrei Doncescu

**Affiliations:** Campus de Fouillole, French West Indies University, 97275 Pointe-à-Pitre, France; sebastien.regis@univ-antilles.fr (S.R.); paul.nuiro@univ-antilles.fr (S.P.N.); woody.merat@univ-antilles.fr (W.M.)

**Keywords:** COVID-19 simulation, SIR, fuzzy subsets, multigroup, data-based approach, aggregation operators

## Abstract

**Simple Summary:**

COVID-19 is a rapidly spreading and mutating pandemic. In the case of some people the disease can be fatal It has been observed that weight and age are parameters of comorbidity If it is difficult to take into account the uncertainties related to the combination of the two parameters to build up a model of simulation. Therefore, we propose in this article a SIR/SIH model with fuzzy parameters which allows us to simulate the pandemic in the absence of barrier actions and vaccines.

**Abstract:**

In this paper, we propose a multi-group SIR to simulate the spread of COVID-19 in an island context. The multi-group aspect enables us to modelize transmissions of the virus between non-vaccinated individuals within an age group as well as between different age groups. In addition, fuzzy subsets and aggregation operators are used to account for the increased risks associated with age and obesity within these different groups. From a conceptual point of view, the model emphasizes the notion of Hospitalization which is the major stake of this pandemic by replacing the compartment R (Removed) by compartment H (Hospitalization). The experimental results were carried out using medical and demographic data from the archipelago, Guadeloupe (French West Indies) in the Caribbean. These results show that without the respect of barrier gestures, a first wave would concern the elderly then a second the adults and the young people, which conforms to the real data.

## 1. Introduction

COVID-19 is a pandemic that has taken by surprises with the speed of its expansion and by the severity of some of its forms. All countries, whatever their level of wealth and development, are overwhelmed by the scale of this phenomenon. This virus with its severe forms as well as its increasingly virulent variants is undermining health systems around the world. The number of deaths and people in intensive care is remaining high which has lead many countries to impose strong health restrictions on their populations: the economic and social consequences are potentially devastating. The biggest challenge for all countries is to manage the flow of patients with serious or critical forms and to manage the hospitalizations of these people in order to avoid a large number of deaths. In parallel with fields of medical research (vaccine, drugs, etc.) simulation and predication approaches help the various institutions to prevent and manage this health, social and economic crisis. In this paper, we present a simulation of COVID-19 in an island context. The peculiarity of this approach is to use a multi-group SIR model and fuzzy subsets. This approach is also a data-based model since it is based on statistical and demographic data from an archipelago in the French West Indies, specifically the island of Guadeloupe. The use of the multi-group approach makes it possible to take the disparities between the age groups into account since the disease takes more severe forms in the elderly compared to adults and young people. The fuzzy subsets reinforce this distinction while bringing nuance in relation to age but also in relation to obesity, which is also an aggravating factor for COVID-19. From a conceptual point of view, the compartment R (Removed) is deliberately replaced by a compartment H (Hospitalization) since this notion of hospitalization is the sensitive point for most countries: such as India or Brazil but also the United States or certain Western European countries. The COVID-19 pandemic is exerting strong pressure on hospital systems, revealing the flaws and weaknesses of these systems, and leading to life and death situations. All the specificities of this approach have a single objective: to allow the simulation to be as close as possible to reality. The paper is organized as follows: after presenting related studies, we will introduce our model and methods, then in the next section, we will provide the experimental results we obtained before concluding.

## 2. Simulation of COVID-19 Using SIR

### 2.1. Related Work

#### 2.1.1. Use of SIR Approach for COVID-19

SIR approach is one of the most frequently used methods for pandemic simulation particularly for COVID-19 [1,2,3,4,5,6] used SIR or its extensions to simulate the spread of COVID-19 among the population in various parts of the world, and these are only examples among many others. Some but not all, of the approaches based on the SIR model incorporate risk factors. Thus, refs. [1,7,8,9] consider age as a risk factor.

In our approach, we incorporate two elements as risk factors, namely age and obesity. The choice of these two risk factors was guided by medical and statistical knowledge derived from real data. Older age is the main risk factor for presenting a severe or critical case among infected people [10,11,12]. Obesity seems to be the second main risk factor [13,14,15]. In particular, the age factor will be taken into account by breaking down the population by age groups.

#### 2.1.2. Multi-Group SIR

To assess the impact of age on the pandemic, the use of the multi-group SIR approach [16,17] is an interesting avenue and makes it possible to create groups by age group. For example, refs. [18,19], or more recently [20,21] for COVID-19, use multi-group SIR to model the spread of disease in different age groups.

Thus, we based our mathematical model on [19], in which the population is subdivided into two age groups. For each *i* = 1, 2



$\left(1\right)\left\{\begin{array}{c} \frac{dS_{i}\left(t\right)}{dt}=-S_{i}\left(t\right)\sum_{j=1}^{2}b_{ij}I_{j}\left(t\right)\\ \frac{dI_{i}\left(t\right)}{dt}=S_{i}\left(t\right)\sum_{j=1}^{2}b_{ij}I_{j}\left(t\right)-y_{i}\left(t\right)I_{i}\left(t\right)\\ \frac{dR_{i}\left(t\right)}{dt}=y_{i}\left(t\right)I_{i}\left(t\right) \end{array}\right.$

with the initial conditions $S_{i}\left(0\right)=S_{i0}\in R_{+},I_{i}\left(0\right)=I_{i0}\in R_{+},R_{i}\left(0\right)=H_{i0}\in R_{+}$, where: $S_{1},S_{2}$: represent the number of susceptible subjects into group 1 and group 2, respectively.$I_{1},I_{2}$: represent the number infectious subjects into group 1 and group 2, respectively.$R_{1},R_{2}$: represent the number of removed people, respectively from group 1 and group 2.$b_{i,j}$: with i,j={1,2} represent the influence of the infectious subject from group *i* on group *j*.$y_{1},y_{2}$ represent the rate of removed people into group 1 and group 2, respectively.We adapted this model in order to use it for 3 age groups: young people, adults and the elderly. In addition to this age differentiation, we also have introduced obesity as an aggravating factor in the SIR model. In order to model these two risk factors linked to the disease, we have used fuzzy subsets that will serve as parameters for the SIR multi-group model equations.

#### 2.1.3. Fuzzy SIR Approach

The use of fuzzy subsets in SIR models [22,23,24] makes it possible to take the uncertainties and inaccuracies inherent in real world data into account, and thus brings a form of realism to the simulation. Fuzzy subsets have already been used for simulations of COVID-19 [25,26] spreading; they make it possible to represent uncertainties related to the biological properties of the virus and infected people, or uncertainties related to the environment. For example, in [26], the fuzzy membership function is linked to the virus-loads, whereas in [25], fuzzy sets are linked to as humidity (promoting the proliferation of the virus) or future policies. In our approach, from real data, we use fuzzy membership functions to model the risk of having a severe form of COVID-19 (requiring hospitalization) as a function of age and as a function of the level of obesity. Demographic data allow us to represent these fuzzy functions from age groups.

## 3. The Fuzzy Multigroup SIR Model

### 3.1. General Presentation

As we said above, the most important challenge for all countries facing COVID-19 is the management of people with severe or critical forms who require hospitalization and which can unfortunately lead to death. From a formal point of view, we first propose to adapt the SIR model by renaming it SIH model, namely Susceptible, Infected and the letter H designates Hospitalized patients who have contracted a severe or critical form. As in [21], we define groups corresponding to the 3 main age groups: young people, adults and the elderly. These 3 groups make sense for the spread and level of severity of the COVID-19 virus since we know that the older we are, the more the level of severity of the virus increases [10,11,12]. Figure 1 provides an idea of the approach for each of the 3 groups.

COVID-19 statistical data [27] and medical data [28] have been used. The fuzzy parameter *y* represents the risk of making a critical form (requiring hospitalization) according to age and to obesity distributed over age.

### 3.2. Mathematical Model

Thus, we start with a mathematical model largely inspired by [19], in which the population is subdivided into three age groups. For each group *i* = 1, …, 3



$\left(2\right)\left\{\begin{array}{c} \frac{dS_{i}\left(t\right)}{dt}=-S_{i}\left(t\right)\sum_{j=1}^{3}b_{ij}I_{j}\left(t\right)\\ \frac{dI_{i}\left(t\right)}{dt}=S_{i}\left(t\right)\sum_{j=1}^{3}b_{ij}I_{j}\left(t\right)-y_{i}\left(t\right)I_{i}\left(t\right)\\ \frac{dH_{i}\left(t\right)}{dt}=y_{i}\left(t\right)I_{i}\left(t\right) \end{array}\right.$

with the initial conditions $S_{i}\left(0\right)=S_{i0}\in R_{+},I_{i}\left(0\right)=I_{i0}\in R_{+},H_{i}\left(0\right)=H_{i0}\in R_{+}$and where: $S_{1},S_{2},S_{3}$: represent the number of young, adult and elderly among potentially ill subjects, respectively. $I_{1},I_{2},I_{3}$: represent the number of young, adult and elderly infectious subjects. $H_{1},H_{2},H_{3}$: the number of young, adult and elderly patients hospitalized. $b_{11},b_{12},b_{13}$: represent respectively the influence of the infectious youth subpopulation of young people, adults and the elderly $b_{21},b_{22},b_{23}$: represent respectively the influence of the subpopulation of infectious adults on young people, adults and the elderly $b_{31},b_{32},b_{33}$: represent respectively the influence of the infectious youth subpopulation of elderly people, adults and the elderly $y_{1},y_{2},y_{3}$ represent the hospitalization rate of young people, adults and the elderly, respectively.

An example of the relationship between two of the three groups is given in Figure 2.

### 3.3. Calculation of Infection Rates b from Real Data

The infection phase is based on infection rates. As this is a multi-group SIR model, there is, therefore, for each group an intragroup infection rate, and intergroup infection rate.

Infection rates are estimated as follows. For each of the three groups, the intragroup infection rate is calculated from the incidence rate measured in age groups. Incidence rate of each group comes from real and actual data [27,29]. The incidence rate is a time series measured at the scale of weeks over the years 2020 and 2021 (see Figure 3). Like the infection rate, the intragroup infection rate will be a time series whose values will be proportional to those of the rates of incidence of each group.

As for the intragroup infection rates, the intergroup infection rates are calculated from evaluations of real clusters in the population.

Thus, the intergroup infection rates between adults–young people, young–adults, adults–elderly, elderly–adults, young–elderly and elderly–young people are calculated from the number of infected persons in clusters whose context is the extended family [27].

### 3.4. The Fuzzy Subsets and Fuzzy Parameter y

After the infection phase, we will look at the number of hospitalization. Hospitalizations due to COVID-19 (whether in a normal bed or in an intensive care unit) are caused by critical forms of the disease. However, age and obesity are aggravating factors which lead to critical forms. This is why these two parameters are integrated in Fuzzy-SIR model as fuzzy subsets.

As a reminder, a classical subset *A* of *X* is defined by a characteristic function which takes the value 0 for the elements of *X* not belonging to *A* and 1 for those which belong to *A* (see Equation (Equation 1)):(1)$$\chi_{A}\;:X\to \left\{0,1\right\}$$ A fuzzy subset *B* of *X* is defined by a membership function which associates to each element *x* of *X* a degree *f (x)* between 0 and 1, with which *x* belongs to *B* (see Equation (Equation 2)).
(2)$$f_{B}\;:X\to \left[0,1\right]$$

As stated above, older age is the main risk factor for the severe or critical cases and obesity is the second main risk factor. Generally, in demographic data, the population is divided into three age groups: young people, adults and the elderly. The severity of COVID-19 disease roughly corresponds to these three groups [10,11,12]: young people who are infected often have a mild form of the disease, adults generally have a more serious form and correspond to the group of severe patients, and elderly people represent the majority of critical cases. The age membership function presented in Figure 4 shows that people between 30 and 60 years old are more vulnerable than the young people but less than the older group. Secondly, the membership function between 30 and 60 has been chosen linear. This fuzzy membership function is not specific only to the islands of Guadeloupe but can be generalized anywhere in the world and it depends on the observations made.

Of course, the second risk factor, obesity, has a membership function similar to the age, but we made a difference between women and men [28]. Obesity risk can lead to the critical form of COVID-19, even for young people [13,14,15].

The medical data [28] for Guadeloupe archipelago provided information on the distribution of obesity by age and sex. We model this risk of critical form linked to obesity (leading to hospitalization) as a membership function which considers also the age for men and for women. The proposed fuzzyfication is presented in Figure 5.

According to the medical data, the proportion of obese women is about double that of men among adults, which implies a greater risk. The proportion of obese young people is low for both sexes. Concerning the elderly, majority of individuals are obese, regardless of gender. This aspect increases the risk and explains the shape of the membership functions described in the figure above.

From these two membership functions of men and women for obesity, we considered a new observation obtained by aggregating the values of these two functions. According to our data related to the ratio between men and women in hospital due to COVID-19 [27,30,31], we defined some weights that ponder these two membership functions. In order to respect the ratio of hospitalizations due to COVID-19 in Guadeloupe [27], we use the following weighted average:(3)$$0.6\times f_{men}+0.4\times f_{women}$$
where $f_{men}$ is the fuzzy membership function for obesity of men and $f_{women}$ is the fuzzy membership function for obesity of women.

We thus have only one value for the risk of hospitalization in relation to obesity.

### 3.5. Fusion of Membership Functions

To estimate the hospitalization rate, we have merged the two fuzzy functions corresponding respectively to the risk associated with age and the risk associated with obesity. The merger is carried out with different types of aggregation operators:the arithmetic meanthe triple Π operator [32]

These two operators belong to two different categories [33]:the arithmetic mean is a averaging operator: these operators provide a value between the maximum and the minimum. It is a compromise.the triple Π operator is a full reinforced operator [33,34]: these operators are both positively reinforced and negatively reinforced. They favor extremes and accentuate differentiation.

In other words, the mean models a moderation and a balance of obesity and age factors, while if these two factors have strong values, the triple Π models the worsening of the disease and systematically favors hospitalization.

For each of the three age groups, it is assumed that individuals have the same chance of catching the disease within the group. Therefore, we will model, by a uniform distribution, the probability of catching a form of COVID-19 involving hospitalization. Thus, via the continuity of the fuzzy membership functions (respectively for age and obesity), we can simulate the values to be used for the hospitalization rates. Each of the two values is then recovered and merged using one of the two aggregation operators. This result of the fusion then represents the hospitalization rate $y_{i}$ for each of the three age groups.

## 4. Results

We used the Euler method to solve the system of Equation (Equation 2), the estimated data of confirmed coronavirus cases presented in [27] and the following initial conditions presented in Table 1. In the following results, $S_{1}$, $I_{1}$ and $H_{1}$ correspond to the proportions of Susceptible, Infected and Hospitalized people among young people. Likewise, $S_{2}$, $I_{2}$, $H_{2}$ represent the proportions of Susceptible, Infected and Hospitalized people in adults, and $S_{3}$, $I_{3}$, $H_{3}$ represent the proportions of Susceptible, Infected and Hospitalized people in the elderly. In Table 2, infection and hospitalization rates are presented and described. Values for infection rates are based on actual data which is normalized, while values for hospitalization rates are based on merging fuzzy membership functions.

We used Maple on a computer with a AMD RYZEN 7 processor at 3.6 GHz and 8 GB of RAM to do simulations. In the following lines, we present in the form of graphs, the results obtained by running simulations over approximately 300 days.

In Figure 6, the peak of the infection appears around day 150, i.e., at the end of the containment in Guadeloupe and in the rest of France, which took place on 11 May 2020 (remember that in this simulation there is no formal consideration of barrier gestures or social distancing). This peak in infections is rapid and reflects a sudden explosion of COVID-19 cases in young people. The curve of hospitalizations shows an exponential growth, but this is lower than the growth of infections, since young people are less affected by the severe form of COVID-19. It is recalled that in this model, there is no compartment for discharge from hospital.

In Figure 7, the pick of infection appears at the same time as that of young people, around day 150. At this peak, the curve of the infected and that of the hospitalized have the same growth: this is due to the fact that adults more easily develop severe or severe forms due to their immunodeficiency (age and obesity). On the curve of the infected, at the level of this peak (which represents the first wave of patients), there is a slight decrease followed by an increase as high as the first. A second wave appears at the time of the day 230: compared to the real data, we can compare this second wave with that which was actually observed toward the end of August at the beginning of September 2020 (see [29]).

In Figure 8, the hospitalization curve grows significantly faster than the infection curve, because this age group directly develops severe forms of the disease given their predisposition (age) and the large number of obese and overweight people. in this age group.

In Figure 9, shows that during the first wave (first peak) young people are the first to be affected, then adults and finally the elderly. It could be argued that it was young people who initiated the transmission of the virus to older groups. The second wave of the elderly comes after that of adults and we can conjecture that this wave is therefore the consequence of the wave that appeared in adults a few days before and which was transmitted by intergenerational mixing (via the intergroup infection rate).

In Figure 10, looking at the trends in the curves, we see that adults were the first affected, certainly because they are the most active, and therefore the most exposed in the population (most often via the intragroup infection rate). Older people follow because they are the most vulnerable people (due to their immune deficiency).

In Figure 11, the curves of the infections in the three age groups are calculated with the 3Π as the aggregation operator. The results are close to those obtained with the arithmetic mean (see Figure 9). However, we can see some differences in the spikes present on the contamination waves. This can be explained by the fact that the triple Pi is more sensitive and a priory produces a greater variability, which is often more in line with reality.

In Table 3, the data from our model and the real data are compared. Here, we see that for the groups of elderly people, the proportion of people hospitalized is quite close to the real data, especially when comparing the data from the model (taken at the start of the peak) to data from the second wave (between August and September 2020), the largest observed in Guadeloupe to date. This is also the case for the adult group. Taking the immunodeficiencies (age and obesity) of these two age groups into account in the model can explain the proximity of the results of the model to the real data. the percentage of young people hospitalized in our model is greater than that of the real data; we can assume that this difference is due to the failure to take barrier gestures into account in our model.

## 5. Conclusions and Perspectives

In this paper, we have proposed a model of the spreading of COVID-19 in an insular context, namely the archipelago of the Guadeloupe F.W.I. Our main contribution is to show the benefits of using a multigroup SIR model, using fuzzy inference. The data used in this model are the real data from the pandemic in the Guadeloupe archipelago. From a conceptual point of view, the compartment R (Removed) has been voluntarily replaced by compartment H (Hospitalization). We have done so because the notion of hospitalization is the most important issue for most countries.

The plasticity of this model (through fuzzy sets and aggregation operators) makes it easier to take into account the uncertainties concerning the major risk factors (age, obesity, and gender). This analytical mode, being without time delays and including intergenerational mixing through the intergroup rates, is well suited to describe the real situation of Guadeloupe.

Nonetheless, there is a significant gap between the results obtained in our simulation and those of reality. As indicated this can be explained by the absence of barrier gestures, social distances and vaccination. The working hypothesis used in our model, namely of not leaving the hospital compartment, after infection, may also be a factor. The results show that the trend is towards a consequent increase in hospitalization. Preventative and/or corrective measures at this level should be considered. Future work will focus on also taking into account the addition of compartment modeling discharges from hospitalization (either death or recovery) and sanitary measures (wearing a mask, social distancing, and vaccination) into account.

## Figures and Tables

**Figure 1 biology-10-00991-f001:**
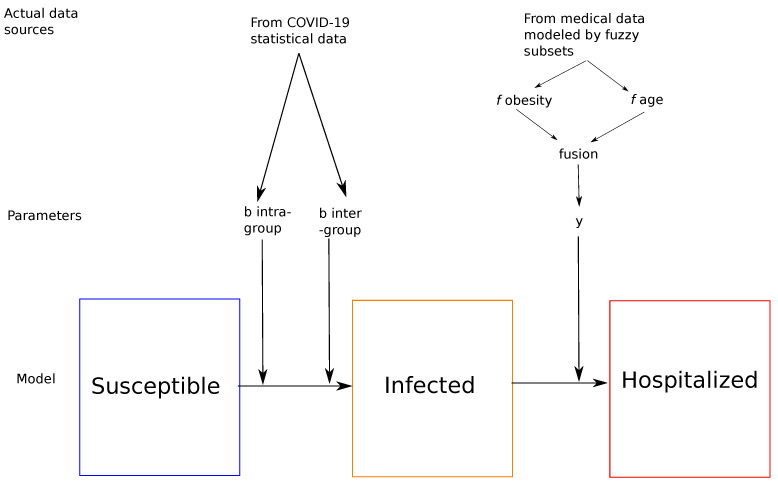
Representation of the model of each age group with the relationship between compartments.

**Figure 2 biology-10-00991-f002:**
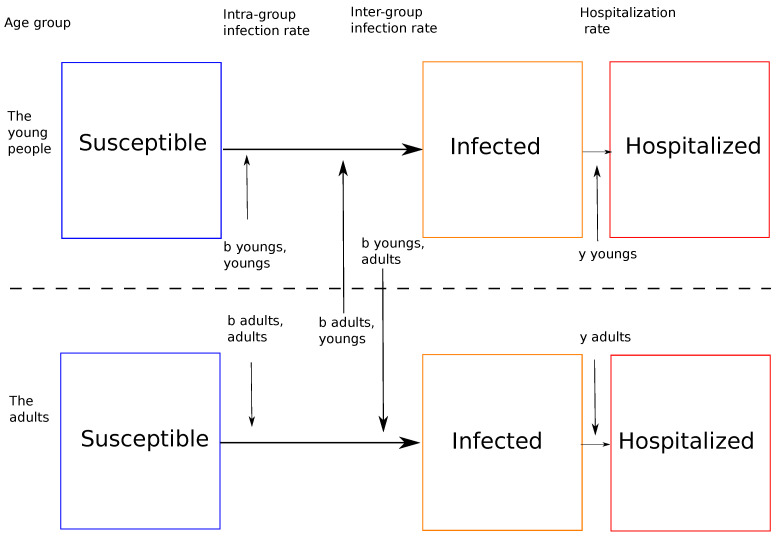
Example of relationship between the group of young people and that of adults. The rates of intergroup infections reflect the contact between these two groups.

**Figure 3 biology-10-00991-f003:**
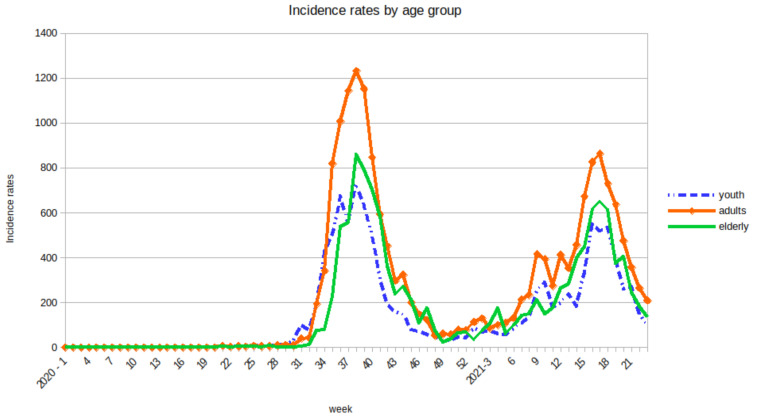
Incidence rates of COVID-19 per 100,000 inhabitants for each age group in Guadeloupe. The rates are given for the 52 weeks of the year 2020 and the first 23 weeks of the year 2021 source: [29].

**Figure 4 biology-10-00991-f004:**
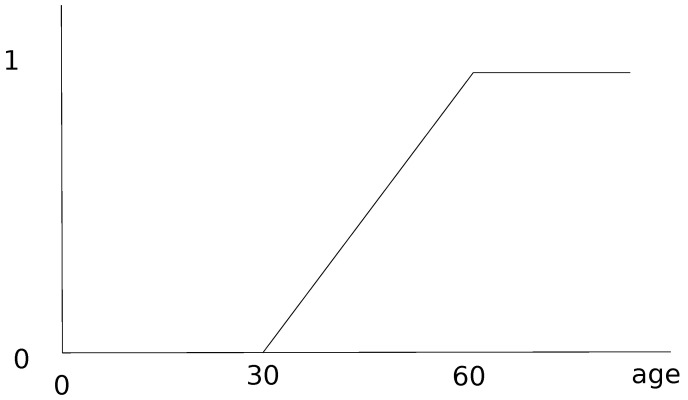
Fuzzy membership function of Hospitalization compartment according to age (in years).

**Figure 5 biology-10-00991-f005:**
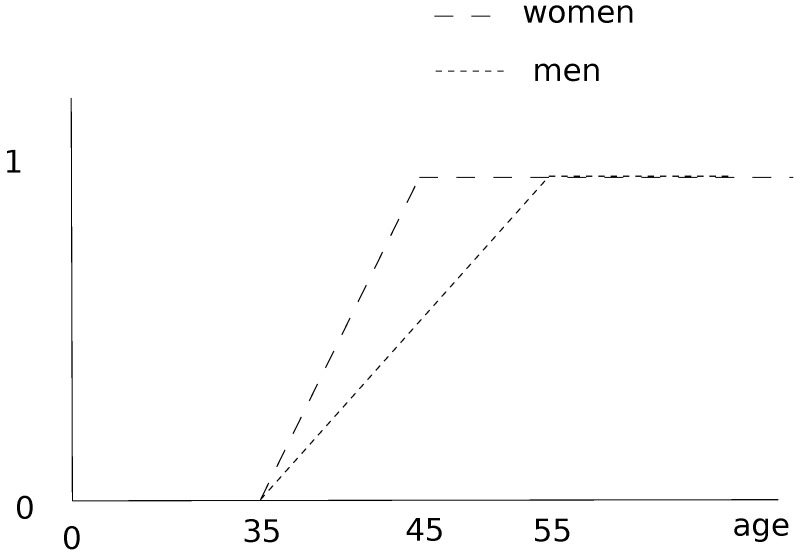
Fuzzy membership function of Hospitalization compartment in terms of obesity for men and women.

**Figure 6 biology-10-00991-f006:**
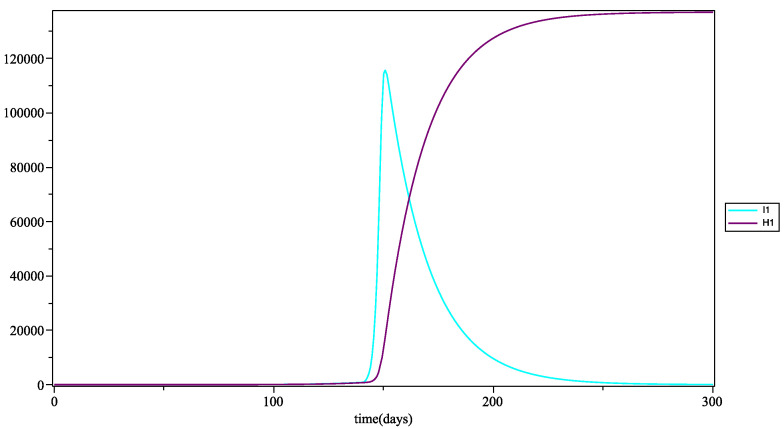
Number of people infected I1 (in blue) *at* time *t*, and number of people hospitalized H1 (in purple) *up to* time *t* for the young group (with the mean as the fuzzy aggregation operator).

**Figure 7 biology-10-00991-f007:**
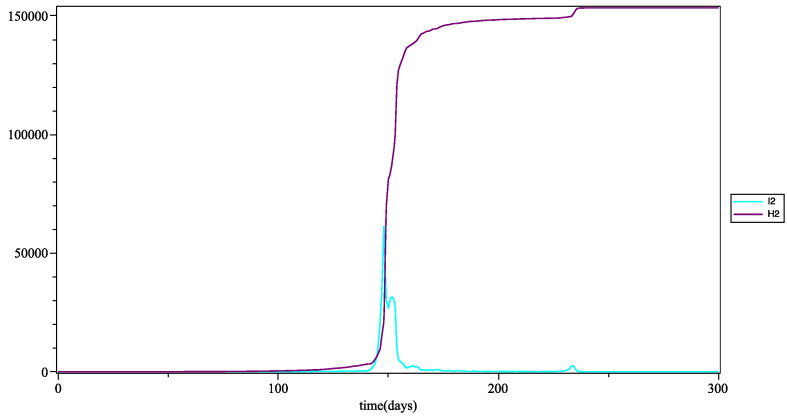
Number of people infected I2 (in blue) *at* time *t*, and number of people hospitalized H2 (in purple) *up to* time *t* for the adult group (with the mean as the fuzzy aggregation operator).

**Figure 8 biology-10-00991-f008:**
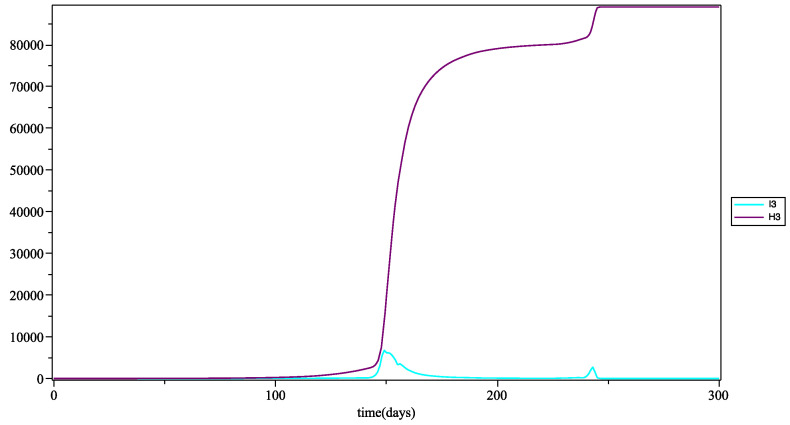
Number of people infected I3 (in blue) *at* time *t*, and number of people hospitalized H3 (in purple) *up to* time *t* for the adult group (with the mean as the fuzzy aggregation operator).

**Figure 9 biology-10-00991-f009:**
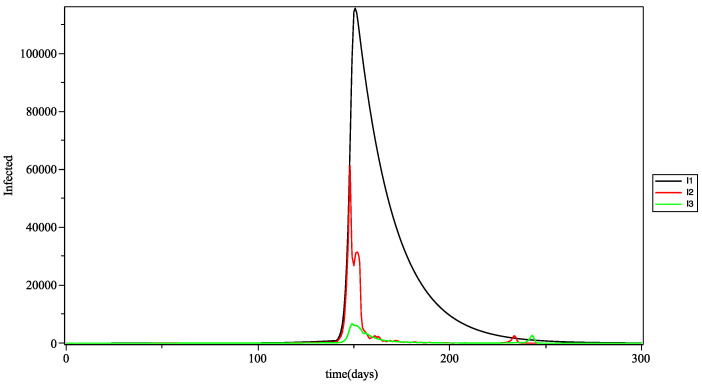
Number of infected people *at* time *t*, for young (I1), adults (I2) and the elderly (I3), with the mean as the fuzzy aggregation operator.

**Figure 10 biology-10-00991-f010:**
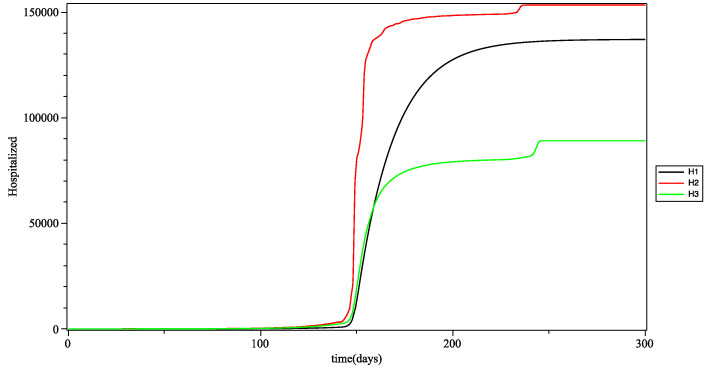
Number of hospitalized people *up to* time *t*, for young (H1), adults (H2) and the elderly (H3), with the mean as the fuzzy aggregation operator.

**Figure 11 biology-10-00991-f011:**
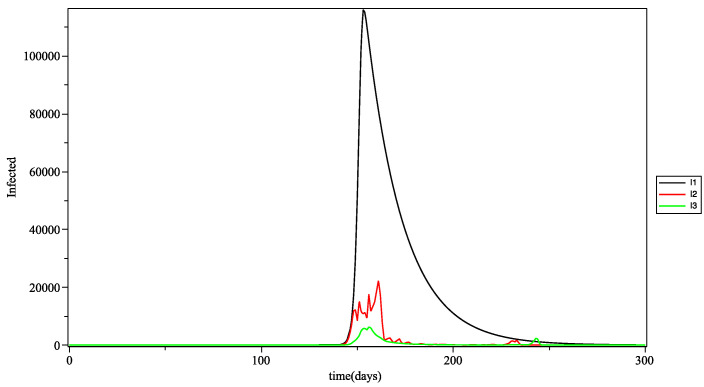
Number of infected people *at* time *t*, for young (I1), adults (I2) and the elderly (I3), with the 3Π as the fuzzy aggregation operator.

**Table 1 biology-10-00991-t001:** Initial values are taken from demographic data source: [35].

Compartment	Initial Value
$S_{1}\left(0\right)$	137,113
$S_{2}\left(0\right)$	153,400
$S_{3}\left(0\right)$	89,197
$I_{1}\left(0\right)$	0
$I_{2}\left(0\right)$	1
$I_{3}\left(0\right)$	0
$H_{1}\left(0\right)$	0
$H_{2}\left(0\right)$	0
$H_{3}\left(0\right)$	0

**Table 2 biology-10-00991-t002:** List of the model parameters used for simulations. *K* and *L* are normalization constants, $r_{i}\left(t\right)$ represents the incidence rate as a time function for the age group *i* [29], and *C* is data on clusters of infected from extended families [27]. For more details, see Appendix A.

Symbol	Description	Calculation of Values
$b_{1,1}$	Infection rate intragroup young	$K\times r_{1}\left(t\right)$
$b_{2,2}$	Infection rate intragroup adults	$K\times r_{2}\left(t\right)$
$b_{3,3}$	Infection rate intragroup elderly	$K\times r_{3}\left(t\right)$
$b_{i,j}$	Infection rate intergroup (i,j)={1,2,3},i≠j	L×C
$y_{i}$	Hospitalization rate for group *i*, (i)={1,2,3}	Fusion of fuzzy values

**Table 3 biology-10-00991-t003:** Comparison of the distribution (in percentage) of hospitalizations in the age groups for the simulation and the real data at day 140 and 248 ([36]).

% for Age Group	Simulation at Day 140	Real Data at Day 140	Real Data at Day 248
youth	18.5%	3.4%	8%
adults	29.4%	31%	45%
elderly	52.1%	65.6%	47%

## Data Availability

Data and samples of the compounds are available from the authors.

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
