# Peer review of "A Data-Based Approach Using a Multi-Group SIR Model with Fuzzy Subsets: Application to the COVID-19 Simulation in the Islands of Guadeloupe"

_biology, 2021, doi:10.3390/biology10100991_

Round 1

Reviewer 1 Report

The paper is topical and relevant and its main purpose, I believe, must be to provide valuable direction to health care planners in Guadaloupe.  However, in its present form I cannot see this paper providing the clear advice such planners require.

I initially liked the concept of replacing the removed category in the standard SIR model with hospitalization as one compartment /category of the modelling. I have long felt that measuring deaths and case numbers was obscuring hospitalizations  that are the pinch point in this whole pandemic episode. However, we cannot end up with everyone moving into hospital, which is how I interpret many of the results, for instance figures 6, 7, 8 etc.

The model predicts a strong emphasis on infection in young people.  Is this because there are so many young people in the island? How does this relate to other island communities and other societies? Surely we need to know this variable – we need to know more about the population in which the virus is transmitting. The conclusions are insufficiently explicit to tell me what is actually happening. The model predicts more than five times the number of hospitalizations in young people than shown by real data at d140. If this was new cases or infections, there might be a plausible explanation, but these are predicted hospitalizations and far from accurate. I would want to see how this inaccuracy can be reduced or removed before putting the data and results to health planners.  I cannot see here the results of including obesity, which was included as an important risk factor in the methodology and overall concept for this work, and its effect on hospitalizations. What happened here?

I would consider rereading a thoughtful redraft of this paper filling in gaps, such as those mentioned but also others  It should be written in such a way that a medical service senior manager can take away the main messages he or she needs for planning bed numbers and staff requirements etc., which as I mentioned in my opening sentence, must be the point of the work. I found the English text very well written but there are a number of minor errors that need to be corrected and edited.  

Author Response

Manuscript ID

biology-1332577

Reviewer 1

The paper is topical and relevant and its main purpose, I believe, must be to provide valuable direction to health care planners in Guadeloupe.  However, in its present form I cannot see this paper providing the clear advice such planners require.

  • I initially liked the concept of replacing the removed category in the standard SIR model with hospitalization as one compartment /category of the modelling. I have long felt that measuring deaths and case numbers was obscuring hospitalizations  that are the pinch point in this whole pandemic episode. However, we cannot end up with everyone moving into hospital, which is how I interpret many of the results, for instance figures 6, 7, 8 etc.
  • The model predicts a strong emphasis on infection in young people.  Is this because there are so many young people in the island? How does this relate to other island communities and other societies?
  • Surely we need to know this variable – we need to know more about the population in which the virus is transmitting. The conclusions are insufficiently explicit to tell me what is actually happening.
  • The model predicts more than five times the number of hospitalizations in young people than shown by real data at d140. If this was new cases or infections, there might be a plausible explanation, but these are predicted hospitalizations and far from accurate. I would want to see how this inaccuracy can be reduced or removed before putting the data and results to health planners.
  • I cannot see here the results of including obesity, which was included as an important risk factor in the methodology and overall concept for this work, and its effect on hospitalizations. What happened here?
  • I would consider rereading a thoughtful redraft of this paper filling in gaps, such as those mentioned but also others  It should be written in such a way that a medical service senior manager can take away the main messages he or she needs for planning bed numbers and staff requirements etc., which as I mentioned in my opening sentence, must be the point of the work. I found the English text very well written but there are a number of minor errors that need to be corrected and edited.  

Responses from Authors

------------------------------------------------------------------------------------------------

  • The results of the figures on infections and then on hospitalizations are mainly due to the fact, as we have already pointed out, that nobody respects the barrier gestures and non-vaccination in this simulation. This is a scenario that could be considered catastrophic, but it helps explain the usefulness of social distancing and see how the hospital system could explode as we point out in the introduction (speaking of India or Brazil). However, the absence of barrier gestures automatically creates a gap in relation to the real data.

  • The arrival of a large number of young people stems above all from the fact once again that the barrier gestures and vaccination are not respected in real lige and in the simulation as is underlined in lines 253 to 255: "On the other hand, the percentage of young people hospitalized in our model is greater than that of
    the real data: we can assume that this difference is due to the failure to take into account vaccination in our model."

  • To create this multi-group SIR model, we are based on the only data available in Guadeloupe and France: figures and data according to certain age groups. The other possible factors (notably ethnic origin) are not authorized or available in France .

  • The difference in the number of hospitalized patients between this simulation and the actual figures is mainly due, in our opinion, to the absence of barrier gestures in the simulation. As a result, slowly but surely, the infected slip into the hospital compartment. In our perspectives written in the conclusion, we indicate that we intend to work on compartments modeling the discharge from hospitalization (death or recovery), and sanitary measures.

  • The inclusion of obesity in the simulation was modeled and integrated into the hospitalization rate allowing passage from the infected compartment to the hospitalized compartment. It therefore directly influences the speed of passage between the infected and the hospitalized. This is particularly emphasized respectively for the group of seniors and adults respectively on lines 218 to 220 and lines 226 to 228.

  • Indeed, for a medical service senior manager, this article would be incomplete because it simulates a situation without barrier gesture and does not take into account the discharge of hospital patients.
    However, our work presented in this article lays conceptual and practical bases (introduction of the hospitalization compartment, consideration of obesity in the form of a fuzzy subset in a SIR model, differentiation of age groups in a multi-group SIR) which seems to us to be intrinsically important. The combination of these conceptual and practical bases is, a priori, not insignificant contributions of this article.

The points underlined by the reviewer (difference in figures from reality) are mainly taken into account in the modeling envisaged in perspective (we are currently working on it even if obstacles persist). We modify our conclusion and will attempt to emphasize these perspectives even more in this conclusion.

Reviewer 2 Report

  1. The author should give the reasonability of the parameter values in Table 2.
  2. There are some typos, e.g., line 199, [ref].
  3. What are the suggestions of this article to control the spread of COVID-19?

Author Response

Manuscript ID

biology-1332577

Reviewer 2

  1. The author should give the reasonability of the parameter values in Table 2.
  2. There are some typos, e.g., line 199, [ref].
  3. What are the suggestions of this article to control the spread of COVID-19?

--------------------------------------------------------------------------------------------------------------------------------------

Responses from Authors

  1. The author should give the reasonability of the parameter values in Table 2.

Did it !

---------------------------------------------------------------------------------------------------------------------------------

  1. There are some typos, e.g., line 199, [ref].

Did it !

---------------------------------------------------------------------------------------------------------------------------------

  1. What are the suggestions of this article to control the spread of COVID-19?

Answer:

In fact, the main objective of this article is above all methodological and focuses on 3 points:

  • the introduction of the concept of the hospitalization compartment in the SIR model
  • the introduction of fuzzy subsets with as main contribution the use of aggregation operators to take into account comorbidities (age, obesity) and non vaccination.
  • the application of a multi-group method to consider the demography of the area studied (here Guadeloupe)

This model considers three parameters: age, obesity, and non-vaccination. Today Guadeloupe is in a big distress due to the characteristics mentioned and we have the strong conviction it could happened in other regions of the world.

However, if this approach can help policymakers to better anticipate the crisis, then that would be satisfactory. In the summary and on line 211 and 255, we explain that the barrier gestures and non-vaccination are not respected, is going to high rate of infected and hospitalized people.

Reviewer 3 Report

The paper is well written and the work is interesting. The authors propose a multi-group SIR model in order to simulate the spread of COVID-19 in an island of Guadeloupe. The fuzzy multi-group SIR model is investigated using a numerical approach namely the Euler method. The experimental results sound well. I think this paper can be accepted after some minior revision in English, the authors should check the paper carefully, such as in page 7, "In this numerical approach, we used the Euler method to solve the system of equations ( 2 ) , the estimated data of confirmed coronavirus cases presented in [ref]? and the following initial conditions presented in table1. " The errors "[ref]?" should be careful and "table1" should be "table 1" .

Author Response

Manuscript ID

biology-1332577

Reviewer 3

The paper is well written and the work is interesting. The authors propose a multi-group SIR model in order to simulate the spread of COVID-19 in an island of Guadeloupe. The fuzzy multi-group SIR model is investigated using a numerical approach namely the Euler method. The experimental results sound well. I think this paper can be accepted after some minor revision in English, the authors should check the paper carefully, such as in page 7, "In this numerical approach, we used the Euler method to solve the system of equations ( 2 ) , the estimated data of confirmed coronavirus cases presented in [ref]? and the following initial conditions presented in table1. " The errors "[ref]?" should be careful and "table1" should be "table 1" .

--------------------------------------------------------------------------------------------------------------------------------------

Responses from Authors

We would like to thank for the remarks of the Reviewer and a new version has been uploaded considering :

"In this numerical approach, we used the Euler method to solve the system of equations ( 2 ) , the estimated data of confirmed coronavirus cases presented in [ref]? and the following initial conditions presented in table1. " The errors "[ref]?" should be careful and "table1" should be "table 1" .

---------------------------------------------------------------------------------------------------------------------------------------

Thank you !

Round 2

Reviewer 2 Report

This revision is ok now.